# An Immunconjugate Vaccine Alters Distribution and Reduces the Antinociceptive, Behavioral and Physiological Effects of Fentanyl in Male and Female Rats

**DOI:** 10.3390/pharmaceutics14112290

**Published:** 2022-10-26

**Authors:** Colin N. Haile, Miah D. Baker, Sergio A. Sanchez, Carlos A. Lopez Arteaga, Anantha L. Duddupudi, Gregory D. Cuny, Elizabeth B. Norton, Thomas R. Kosten, Therese A. Kosten

**Affiliations:** 1Department of Psychology & TIMES, University of Houston, Houston, TX 77204, USA; 2Department of Pharmacological & Pharmaceutical Sciences, University of Houston, Houston, TX 77204, USA; 3Department of Microbiology & Immunology, Tulane University School of Medicine, New Orleans, LA 70112, USA; 4Menninger Department of Psychiatry, Baylor College of Medicine, Houston, TX 77030, USA; 5The Michael E DeBakey Veteran’s Affairs Medical Center, Houston, TX 77030, USA

**Keywords:** opioids, analgesia, overdose, vaccines, conjugate, adjuvant, antibodies, anti-fentanyl vaccine

## Abstract

Fentanyl (FEN) is a potent synthetic opioid associated with increasing incidence of opioid use disorder (OUD) and fatal opioid overdose. Vaccine immunotherapy for FEN-associated disorders may be a viable therapeutic strategy. Here, we expand and confirm our previous study in mice showing immunological and antinociception efficacy of our FEN vaccine administered with the adjuvant dmLT. In this study, immunized male and female rats produced significant levels of anti-FEN antibodies that were highly effective at neutralizing FEN–induced antinociception in the tail flick assay and hot plate assays. The vaccine also decreased FEN brain levels following drug administration. Immunization blocked FEN-induced, but not morphine-induced, rate-disrupting effects on schedule-controlled responding. Vaccination prevented decreases on physiological measures (oxygen saturation, heart rate) and reduction in overall activity following FEN administration in male rats. The impact of FEN on these measures was greater in unvaccinated male rats compared to unvaccinated female rats. Cross-reactivity assays showed anti-FEN antibodies bound to FEN and sufentanil but not to morphine, methadone, buprenorphine, or oxycodone. These data support further clinical development of this vaccine to address OUD in humans.

## 1. Introduction

Illicit synthetic opioid use has led to increases in opioid use disorder (OUD) and accidental opioid-related overdose deaths creating a significant public health crisis. Over a period of 12 months, (2019–2020) overdose deaths in the United States increased to the highest ever recorded (81,000) and alarmingly, the latest statistics are even higher with nearly 70% of all overdose deaths (96,700) involving opioids [1,2]. Also concerning is that overdose deaths have dramatically increased in young people (age 10–24 years). Most recent estimates indicate the years of life lost secondary to unintentional drug overdose over a 5-year period (2015–2019) is greater than 1.25 million years [3].

Fentanyl (FEN) is a synthetic opioid agonist that is approximately 100 times more potent than morphine [4]. It is decidedly lipophilic and rapidly enters the central nervous system activating mesocorticolimbic circuitry producing highly reinforcing euphoria [5,6,7]. A significant rise in overdose deaths beginning in the early 2000’s showed that FEN and its derivatives adulterated other misused substances that led to increased lethality [8] and, that between 2010–2016, over 50% of stimulant-related overdose deaths involved this opioid [9]. FEN is also found in counterfeit benzodiazepines, oxycodone and hydrocodone/acetaminophen pills increasing the chances of fatal overdose in individuals who do not ordinarily consume opioids [8,10]. In addition, during the SARS-CoV-2 pandemic, individuals with OUD were more prone to poor outcomes from COVID-19 infection and overdose [11,12].

Current treatments for OUD are methadone, buprenorphine and naltrexone, and their effectiveness depends upon formulation (e.g., extended release, depot, implantable), compliance, access to medications, and the mis-used opioid. FEN use and overdose is a particular treatment challenge that is not adequately addressed with current medications because of its pharmacodynamics and managing acute overdose with the short-acting antagonist naloxone is not appropriately effective particularly with typical intra-nasal administration [13]. Multiple doses of naloxone are often needed to reverse FEN’s fatal effects (e.g., respiratory depression, truncal rigidity or “wooden chest syndrome”) [14,15]. This pharmacodynamic efficacy challenge can be addressed using immunotherapies that prevent FEN from entering the brain preemptively circumventing its reinforcing and overdose effects [16].

OUD is a chronic disorder that even with maintenance therapy has relapse rates approaching 90% largely due to poor medication compliance with current agents [17,18]. Immunotherapy using vaccines can reduce this compliance problem by requiring less frequent administration. Vaccine immunotherapy targets small molecule antigens such as opioid compounds consisting of the antigen (hapten) linked to an immunogenic carrier protein that stimulates the immune system to generate antibodies. An adjuvant is often added to the formulation to increase immunogenicity (antibody formation) against the hapten. After vaccination, when an opioid is consumed, antibodies bind to the target opioid in the periphery and prevent it from getting into the brain or affecting other organs. The kidneys subsequently eliminate bound drug from the body. Numerous pre-clinical studies targeting a broad range of opioids have demonstrated vaccine efficacy [16,19,20,21,22].

Recent studies testing various formulations of anti-FEN vaccines show efficacy in attenuating the behavioral and lethal effects of FEN and its derivatives [23,24,25] including findings from our group [26]. Our vaccine formulation is composed of a FEN-like hapten containing a linker with a carboxyl moiety suitable for carbodiimide coupling chemistry to form amide bonds to lysine residues on the carrier protein CRM197, a genetically deactivated diphtheria toxin contained in several FDA-approved conjugate vaccines. We combined our conjugate (FEN-CRM) with an adjuvant derived from heat-labile enterotoxins from *E. coli* (LT) named dmLT or LT(R192G/L211A) that has been combined with other vaccines in several human clinical trials [27,28,29].

Although the prevalence of OUD and overdose is higher among males this trend appears to be narrowing most notably among certain age groups [30]. Both pre-clinical and clinical studies have long neglected to include female subjects largely due to the potential impact of hormonal fluctuation. There is however, other important gender and sex differences that relate to pharmacokinetics and pharmacodynamics of opioids that may contribute to divergent effects on drug reinforcement brain circuitry and areas regulating opioid-induced antinociception and withdrawal [31,32,33,34]. A large literature has repeatedly shown that male rats are more sensitive to the antinociceptive effects of opioids [35] although other studies find no sex-differences in FEN-induced analgesia [36] or metabolism [37]. Yet, compared to male rats, female rats self-administer more FEN and are more sensitive to certain behavioral effects of the drug [38,39,40]. Further, female rats typically develop higher levels of antibodies following vaccination compared to male rats consistent with results from human clinical trials [41]. Based on evidence indicating sex differences in opioid-induced antinociception and vaccine responses, we chose to test our vaccine in both male and female rats. To our knowledge, there have been no studies directly comparing the immunogenicity and efficacy of a vaccine developed for a substance use disorder in both male and female Sprague Dawley rats.

We previously showed that FEN-CRM+dmLT generated significant amounts of anti-FEN antibodies that were associated with significant blockade of FEN’s analgesic effects in mice. The current study was conducted to extend the assessment of FEN-CRM+dmLT to male and female rats and expand upon the behavioral tests of nociception to determine if it also blocks FEN-induced disruption of schedule-controlled responding and depression of physiological measures associated with overdose. We also sought to show anti-FEN antibody specificity and the ability of the vaccine to prevent FEN from entering the brain.

## 2. Materials and Methods

### 2.1. Animals

Sixty Sprague Dawley rats (Charles River, Willmington, MA, USA, male N = 30, female N = 30) were utilized to assess immunogenicity of the vaccine formulation, nociception, and FEN brain levels. Thirty-two rats (male N = 16, female N = 16) were used for the schedule-controlled responding experiments and twenty-eight rats (male N = 13, female N = 15) for the physiology experiment. The vivarium was maintained on a 12:12 light/dark cycle (lights on at 7:00 AM). Animals had ad libitum access to food and water however, for the schedule-controlled responding study, food was restricted during the week. The University of Houston Institutional Animal Care and Use Committee approved the animal protocol in accordance with guidelines set forth in the ‘Guide for the Care and Use of Laboratory Animals 8th Edition’.

### 2.2. Drugs

FEN was purchased from Cayman Chemical (Ann Arbor, MI, USA) and morphine from Sigma Aldrich (St Louis, MO, USA). Drugs were dissolved in sterile saline and administered (mg/kg) subcutaneously 30-m prior to testing or obtaining serum and brain samples.

### 2.3. Antigens and Adjuvants

FEN-BSA, morphine-BSA, methadone-BSA, buprenorphine-BSA and oxycodone-BSA were purchased from Cal BioReagents. SUFENTANIL-BSA was synthesized and extensively characterized in our laboratory (see Appendix A). SUFENTANIL-CRM conjugate characterization by MALDI-TOF gave a haptenization ratio of 2.76. FEN-CRM synthesis has been described previously [26,42]. Briefly, FEN-CRM was synthesized using a FEN derivative with a carboxylic acid linker coupled to lysine residues on CRM197 (Fina Biosolutions, Rockville, MD, USA). The FEN hapten was created in a series of four chemical reactions. The product of each step was characterized and validated by ^1^H and ^13^C NMR spectrum and purity of the FEN hapten was validated by HPLC. The final product was then conjugated to CRM197 as described in detail [26]. FEN-CRM conjugate characterization by MALDI-TOF gave a haptenization ratio of 2.4–2.6. Prior to immunization, the conjugate was dialyzed in PBS (Slide-A-Lyzer, Thermo Scientific, Waltham, MA, USA), sterilized by passing it through a 0.2 μm filter (Acrodisk, Pall Corporation, Ann Arbor, MI, USA) and quantified using a protein assay kit (BCA, Pierce, Appleton, WI, USA). GLP–grade dmLT was produced according to cyclic GMP (cGMP) specification by IDT in sodium phosphate buffer supplemented with 5% lactose in vials containing 400-μg of lyophilized product in a 3-mL sterile, multidose, Wheaton serum vial and stored at 4 °C. dmLT was re-suspended prior to use with IXPBS. 

### 2.4. Immunization and Sample Collection

The vaccine formulation (5 µg FEN-CRM+1 µg dmLT) was prepared immediately before administration by admixing antigen and adjuvant in sterile PBS in a 100-μL volume per vaccination per animal. Animals were injected with a 0.5 cc insulin syringe into the right and left caudal thigh muscle (50-µL per hind limb) As shown in Figure 1A, immunizations occurred at 0, 3 and 6 weeks. Blood samples were collected from the saphenous vein on weeks 4, 6, 8 and 10 post-initial vaccination. Following administration of the high dose of FEN (0.1 mg/kg) a final collection occurred at week 20 whereby rats were placed under isoflurane anesthesia and the heart exposed via bilateral thoracotomy and blood obtained by cardiac venipuncture. The brain was also removed at this time.

### 2.5. Antibody Levels, Cross-Reactivity and Fentanyl Quantification

Anti-FEN IgG antibody ELISAs were conducted using similar methods described previously [26]. Corning 96 well flat bottom plates (Costar 9018) were coated with 0.2-μg FEN-BSA and detected using AKP-conjugated anti-rat IgG (Sigma). ELISAs were quantified using dilutions of purified rat standard IgG (Sigma) to generate a standard curve that were used to calculate IgG anti-FEN antibody concentrations in samples. Results are expressed as ng/mL. Potential cross-reactivity to various antigens other than FEN-BSA was assessed using the same ELISA methods with the exception that the plate was coated with morphine-BSA, methadone-BSA, buprenorphine-BSA and oxycodone-BSA and serum samples from unvaccinated and vaccinated rats assayed. Concentrations of anti-FEN antibodies for both assays were quantified using four-parameter logistics curve fitting from symmetrical sigmoidal calibrators. Serum and brain FEN concentrations were quantified using ELISA (Fentanyl Direct, Immunalysis, Pamona, CA, USA). Published data showed the ELISA’s sensitivity and specificity for FEN and analogs to be comparable to Liquid chromatography–mass spectrometry [43]. The immunoassay was conducted according to manufacture specifications. Brains were first homogenized in super pure water then underwent centrifugation (10,000 rpm, 10 min) and supernatant removed. Protein quantification of each supernatant from each animal was obtained using BCA protein assay kit (ThermoFisher Scientific). Equal protein concentrations from each rat were assayed for FEN concentration. FEN standards, negative control and samples (serum and brain homogenate) where loaded on the assay plate. Subsequently, 100-µL of Immunalysis enzyme conjugate solution was added. After 60-m of incubation (room temperature, in the dark) wells were washed six times with super pure water. Then, 100-µL of Immunalysis substrate reagent was added and plate incubated for 30-m and 100-µL of stop solution was added to each well to stop the reaction. Absorbance was measured at 450-nm using a plate reader (Multiskan Go, Thermo Scientific). Third order polynomial curve fitting of FEN standard comparators was used to quantify FEN in samples.

### 2.6. Nociception Assays

We assessed FEN nociception using tail flick (Ugo Basile, Collegeville, PA, USA) and hotplate (Columbus Instruments, Columbus, OH, USA) apparatuses. For the tail flick test, heat was applied to the tail 3-cm from the tip. The intensity of the radiant heat was adjusted so that baseline latencies would fall between 2 and 4-s. The time from the onset of the heat to the withdrawal of the tail (latency in seconds) was measured by the apparatus. To avoid tissue damage, the heat stimulus was discontinued after 10-s (cut-off latency). Baseline latencies were obtained three times for each rat in each group after a saline injection and after random administration of one of two doses of FEN (0.05 and 0.1 mg/kg, SC). There was a 1-week washout period between each FEN dose test. For the hotplate test, temperature was set at 53 °C and cut off time was 90-s. Latency for the rat to lick its hind limb (in seconds) was obtained as the primary measure. To control for baseline differences between animals following saline, data are presented as %MPE (percent Maximum Possible Effect) = (test latency − control latency)/(cutoff criterion − control latency) × 100.

### 2.7. Schedule-Controlled Responding

Ten, standard operant chambers (Coulbourn Instruments, Holliston, MA, USA) enclosed in sound-attenuating cubicles (Coulbourn Habitest isolation cubicle) were used for this study. Each chamber was equipped with two levers located on either side of an access area into which a trough was located to dispense food pellets (45 mg; Bio-Serv, Frenchtown, NJ, USA). A house light, a food trough access area light, and two sets of three, colored cue lights, one above each lever, were located within the operant chamber. Stimulus parameters and data tabulation were programmed using Graphic State Notation (version 4.0, Holliston, MA, USA). In short, this experiment assessed the effects of various doses of FEN and morphine (MOR) on schedule-controlled responding for food as we have done previously [44]. Rats were initially trained to lever-press for food pellets under a fixed ratio (FR) 1 schedule of reinforcement in daily 30-m sessions conducted 5 days/week. A total of 50 reinforcers could be obtained by depression of the active lever during daily sessions. Sessions began with the illumination of the house light and both sets of cue lights and terminated after 50 reinforcers were earned or 30-m elapsed, whichever occurred first. The FR requirement was gradually increased until an FR15 was achieved. Test sessions began once the rat showed consistent response rates (<20% variance over two consecutive days) and were conducted twice weekly. The effects of pretreatment (30-m) with randomly administered doses of FEN (0.01–0.1 mg/kg) and MOR (0.3–10 mg/kg) on operant response rates were recorded. Training sessions continued on intervening days. Once rats had been administered all doses of FEN and MOR, they were vaccinated at 0, 3 and 6 weeks (Figure 1A), similar to rats that underwent nociception testing. Following vaccination, rats were retested and again administered FEN and MOR and response rates obtained. Data are presented as percent of baseline (saline administration) response rates.

### 2.8. Physiological Effects

A non-invasive pulse oximetry system (Mouse004F Plus, Starr Life Sciences, Oakmont, PA, USA) was used to measure the ability of our conjugate vaccine to attenuate FEN-induced depression on oxygen saturation, heart rate, and general activity in both male and female rats. Rats were first acclimatized to the sensor that was placed around the dorsal neck for two consecutive days in the testing room. Rats were then habituated to the testing chamber (Coulbourn Instruments, Holliston, MA, USA) for 30-m with the sensor attached to the rat and baseline measures recorded. Following habituation, rats were administered FEN (0.1 mg/kg, SC) and placed back into the testing chamber and measures recorded for 30-m. We chose this time point based on preliminary studies showing near recovery from FEN-induced decreases on oxygen saturation post-FEN administration in both male and female rats.

### 2.9. Statistical Analysis

Anti-FEN antibody levels across weeks were analyzed using two-way ANOVA with Sex as the between group factor with repeated measures on Time. The Geisser-Greenhouse correction for Time was used. Potential differences in %MPE measures (tail flick and hotplate) were analyzed using three-way ANOVA with Vaccine status and Sex as between-group factors and FEN dose (0.05 and 0.1 mg/kg) as a repeated measure. Brain FEN levels across groups were analyzed using two-way ANOVA with Vaccine status and Sex as between-group factors. Schedule-controlled responding data were subjected to a three-way ANOVA representing between group factors of Vaccine status and Sex with Drug dose as repeated measure. Separate analyses by drug (FEN or MOR) were performed. Cross-reactivity data were analyzed with two-way ANOVA representing factors of Vaccine status (serum from CONTROL-male unvaccinated and FEN-CRM-male vaccinated) and coating antigen (FEN-BSA, sufentanil-BSA, morphine-BSA, methadone-BSA, buprenorphine-BSA, oxycodone-BSA). Oxygen saturation, heart rate, and activity counts were analyzed using three-way ANOVA with Vaccine status and Sex as between-group factors with repeated measure on Time using the Geisser-Greenhouse correction. Post hoc Tukey’s or Sidak’s or Newman-Keuls multiple comparisons tests were used to follow up on all significant interactions. GraphPad Prism (v7, San Diego, CA, USA) or Statistica (TIBCO v. 13.2.92.5) was used for statistical analysis with statistical significance defined as *p* < 0.05.

## 3. Results

### 3.1. Anti-FEN Antibody Levels

Anti-FEN IgG antibody levels over ten weeks are shown in Figure 1B. ANOVA yielded significant main effects for Sex (F(1, 28) = 8.604, *p* < 0.01) and Time (F(3, 84) = 13.85, *p* < 0.0001) and a significant Sex X Time interaction (F(3, 84) = 10.37, *p* < 0.0001). For male rats, post hoc multiple comparisons revealed significant differences in anti-FEN IgG antibody levels between 4 and 6 (*p* < 0.01), 8 (*p* < 0.01), and 10 (*p* < 0.05) weeks. For female rats, post hoc multiple comparisons revealed a significant difference in antibody levels between weeks 4 and 6 (*p* < 0.05). Between group comparisons revealed a highly significant difference in antibody concentrations for male rats compared to female rats at weeks 8 (*p* < 0.0001) and 10 (*p* < 0.01). Overall, anti-FEN IgG antibody levels increased over time compared to baseline in male rats and were greater than levels in female rats. 

### 3.2. Antinociception Tests: Tail Flick and Hot Plate

Nociception measures in the tail flick and hot plate assays following a low dose (0.05 mg/kg) and high dose (0.1 mg/kg) of FEN are presented in Figure 2A–D for male rats and in Figure 3A–D for female rats. Analysis (%MPE) of tail flick data revealed significant main effects for Vaccine status (F(1, 56) = 143.03, *p* < 0.0001), Sex (F(1, 28) = 23.51, *p* < 0.0001, FEN dose (F(1, 56) = 23.25, *p* < 0.0001), and significant interactions of Sex X Vaccine status, (F(1,56) = 24.97, *p* < 0.0001), and FEN dose X Vaccine status, (F(1, 56) = 19.77, *p* < 0.0001). Post hoc multiple comparisons showed highly significant differences between male vaccine groups (Control vs. FEN-CRM; *p* < 0.0001) following the low dose of FEN. Analysis of %MPE data following the high dose of FEN showed significant differences between vaccine groups for both male and female rats (*p*’s < 0.0001) indicating that FEN produced robust analgesia that was completely blocked by the vaccine. While the low dose of FEN (0.05 mg/kg) produced significant analgesia in male rats that was completely blocked by the vaccine, this dose did not produce a significant analgesic effect in unvaccinated female rats and thus was not affected by the vaccine. However, the high dose of FEN did produce a modest analgesia in female rats that was blocked by the vaccine.

Results from analysis of %MPE data from the hot plate assay indicated significant main effects for Vaccine (F(1, 56) = 69.43, *p* < 0.0001) and FEN dose (F(1, 56) = 47.77, *p* < 0.0001) and a significant interaction of Vaccine X Dose (F(1, 56) = 41.53, *p* < 0.0001). The main effect of Sex was not significant nor were any of its interaction terms (*p*’s > 0.10). FEN produced a greater analgesic response at the high dose compared to the low dose. Post hoc comparisons did not reveal any significant differences between groups following the low dose of FEN. There were significant differences between vaccine groups following the high dose of FEN for both sexes (*p*’s < 0.05–0.0001).

### 3.3. Fentanyl Brain Levels

Figure 2E and Figure 3E show brain levels of FEN following a high dose administration. ANOVA revealed significant main effects for Vaccine status (F(1, 54) = 599.90, *p* < 0.001) and Sex (F(1, 54) = 10.70, *p* < 0.005) and their interaction (F(1, 54) = 4.53, *p* < 0.05). Post hoc multiple comparisons revealed significant differences between vaccine groups of both sexes (*p*’s < 0.01–0.0001). As expected, greater amounts of FEN were present in the brains of unvaccinated male and female rats compared to vaccinated rats of either sex. Interestingly, female unvaccinated rats had higher brain FEN levels compared to male unvaccinated rats (*p* < 0.01).

### 3.4. Schedule-Controlled Responding

Analysis of the response rates as a percent of baseline following FEN administration from the schedule-controlled responding study are presented in Figure 4A (males) and Figure 4C (females). There were significant main effects for Vaccine status (F(1, 47) = 45.70, *p* < 0.0001) and FEN dose (F(3, 141) = 70.24, *p* < 0.0001) and a significant interaction of Vaccine status X Dose (F(3, 141) = 19.40, *p* < 0.0001). Post hoc comparisons revealed a significant difference between female vaccine groups at the 0.03 mg/kg FEN dose (*p* < 0.05). Following 0.05 and 0.1 mg/kg FEN doses, rates of responding significantly differed between vaccine groups of both sexes (*p*’s < 0.0001). MOR administration effects on lever pressing rates pre- and post-vaccination are presented in Figure 4B (males) and 4D (females). Analysis revealed a significant main effect for MOR dose, (F(3, 135) = 3.96, *p* < 0.01) but no significant main effects of Sex or Vaccine status or of their interaction terms (*p*’s > 0.1). The significant MOR dose effect likely reflects the decrease in response rats seen at the highest dose (10 mg/kg).

### 3.5. Physiology Study

The effects of high dose FEN on oxygen saturation, heart rate, and activity levels in male and female rats are presented in Figure 5A–F. Analysis of oxygen saturation data revealed significant main effects for Sex (F(1, 25) = 5.33, *p* < 0.05), Vaccine status (F(1, 25) = 59.87, *p* < 0.0001), Time (F(6,150) = 19.52, *p* < 0.0001), and significant interactions of Sex X Vaccine status, (F(1,25) = 6.05, *p* < 0.05) and Time X Vaccine status (F(6, 150) = 16.62, *p* < 0.0001). Post hoc comparisons revealed significant differences between male vaccine groups at 5-m (*p* < 0.05), between vaccine groups of both sexes at 10-, 15-, and 20-m, (*p*’s < 0.01–0.0001), and between male vaccine groups at 25-m (*p*’s < 0.01–0.001). FEN decreased oxygen saturation in both male and female unvaccinated rats that was blocked by the vaccine. Analysis of heart rate (bpm) revealed significant main effects for Sex (F(1, 25) = 23.06, *p* < 0.001), Vaccine status (F(1, 25) = 26.74, *p* < 0.001), Time (F(6, 150) = 13.72, *p* < 0.0001), and a significant Time X Vaccine status interaction (F(6, 150) = 6.24, *p* < 0.0001). Post hoc analysis revealed a significant difference between male vaccine groups at 10-m (*p*’s < 0.01), indicating that the vaccine blocked the FEN-induced decrease in heart rate. Analysis of activity counts revealed significant main effects for Sex (F(1, 25) = 113.42, *p* < 0.0001), Vaccine status (F(1, 25) = 36.37, *p* < 0.0001), and Time (F(6, 150) = 6.96, *p* < 0.0001) and significant Sex X Vaccine status (F(1, 25) = 7.55, *p* < 0.05) and Time X Vaccine status (F(6, 150) = 5.43, *p* < 0.0001) interactions. Post hoc comparisons revealed significant differences between male vaccine groups at 20- and 25-m (*p*’s < 0.01). FEN significantly decreased activity counts in male rats and the vaccine blocked this effect. FEN-induced decreases in activity counts were significantly lower in male rats compared to female rats.

### 3.6. Cross-Reactivity Assay

ELISA cross-reactivity assay results are presented in Figure 6. Analysis of anti-FEN antibody binding to various opioid haptens yielded a highly significant main effect for vaccine status (F(1, 167) = 26.33, *p* < 0.0001), coating antigen (F(5, 167) = 11.27, *p* < 0.0001), and a significant interaction (F(5, 167) = 11.52, *p* < 0.0001). Post hoc analysis revealed significant differences between FEN-BSA and all other coating antigens (*p*’s < 0.0001).

## 4. Discussion

The present study expands upon our previous results by further characterizing our anti-FEN vaccine in male and female Sprague Dawley rats. We find that the FEN-CRM+dmLT vaccine: (1) produces anti-FEN antibodies that are long-lasting; (2) blocks FEN-induced nociception in the tail flick and hotplate assays; (3) leads to lower FEN brain levels following FEN administration 20-weeks after initial vaccination; (4) blocks the rate-disrupting effects of FEN, but not MOR, on schedule-controlled responding; (5) protects against FEN-induced decreases in oxygen saturation, heart rate, and activity levels in male rats and decreases in heart rate in female rats; and (6) generates anti-FEN antibodies that bind specifically to FEN and not to other opioids. Additionally, we report on several sex differences in response to the vaccine or to FEN. These findings include: (1) greater anti-FEN antibody levels in male rats compared to female rats across weeks; (2) a more pronounced FEN-induced antinociception in male rats in the tail flick test; (3) greater penetration of FEN into the brain in female rats; and (4) greater FEN-induced decreases on oxygen saturation and heart rate in male rats compared to female rats. These data show our FEN-CRM conjugate vaccine coupled with the adjuvant dmLT can produce anti-FEN antibodies that eliminate the behavioral and physiological effects of FEN. The vaccine formulation includes CRM197, a carrier protein in vaccines already FDA-approved, and GMP-grade dmLT, that has proven safe and effective in many clinical trials [45]. This is the first report to show the effectiveness of this vaccine formulation in male and female rats that extend our previous findings in mice [26] and are consistent with other vaccine studies targeting opioids [22,23,24,25,42,46,47].

We utilized two well-characterized nociception tests—the tail flick and hotplate assays—that are mediated through spinal and supra-spinal pathways, respectively, and provide an index of vaccine effectiveness [48]. Vaccination (IM, at 0, 3 and 6 weeks, Figure 1) with FEN-CRM197+dmLT (5 μg and 1 μg) completely blocked (93–99%) FEN-induced antinociception following two doses of FEN (0.05 mg/kg and 0.1 mg/kg) in the tail flick assay and after the high dose of FEN in the hotplate assay. Interestingly, in our previous study that employed female mice, our vaccine was more effective in the hotplate test compared to the tail flick test across two FEN doses. This may reflect species differences in FEN’s ability to affect spinal vs. supra-spinal pathways or to metabolic differences between species [49,50]. Another group generated a novel FEN-based hapten conjugated to carrier proteins including CRM197 [23]. They showed significant attenuation of FEN’s (0.05 mg/kg and 0.1 mg/kg) analgesic effects in the hot plate test in vaccinated mice (60 µg of conjugate at 0, 14 and 28 days; SC). In that same study, rats vaccinated with 60 µg of conjugate on days 0, 21, 42 and 63 and challenged with FEN (0.075 mg/kg, SC) in the hot plate test showed a significant reduction of FEN antinociception over a 1-h period. Other groups measured the analgesic effects of anti-FEN vaccines using various carrier proteins. For example, KLH (keyhole limpet hemocyanin) conjugated to a FEN-based hapten (25 µg) with 0.5 mg aluminum hydroxide adjuvant administered to mice on days 0, 14 and 28 led to a significant decrease (60% MPE) in the hot plate assay following 0.05 mg/kg (SC) of FEN [25]. Rats vaccinated with the same formulation and challenged with a lower dose of FEN (0.035 mg/kg, SC) also showed a significant reduction in FEN-induced antinociception (93% MPE) in the hot plate test. An extensive series of studies with a FEN hapten conjugated to tetanus toxoid (TT, 10 μg) adjuvanted with liposomes containing monophosphoryl lipid A (20 μg) adsorbed on aluminum hydroxide (30 μg) on weeks 0, 3, 6 and 14 in mice assessed nociception (tail immersion and hotplate) [24]. On week 18, mice were challenged with increasing doses of FEN (0.005–4 mg/kg) and cumulative dose curves showed significant shifts in ED50 values indicating a blockade of FEN-induced antinociception. A final study also utilized the FEN-TT conjugate (50 μg) combined with adjuvants aluminum hydroxide (750 μg) and CpG oligodeoxynucleotide (50 μg) [42]. Mice were vaccinated (IP) at 0, 3 and 4 weeks and tail immersion and hot plate assays were conducted following a range of high FEN doses (0.01–5 mg/kg). Data from both assays indicated several-fold shifts in ED50s demonstrating that this vaccine also attenuated FEN’s antinociception effects. Overall, these studies report that their vaccine produced anti-FEN antibodies that neutralized FEN-induced effects in rodents despite immunization with different anti-FEN formulations that vary in conjugates, adjuvant doses, and vaccination schedules, species, and sex.

Opioid overdose results from acute respiratory depression that rapidly leads to irreversible hypoxic cardiac asystole [51]. This effect is mediated through mu opioid receptors located on medullary inspiratory neurons [52]. We measured oxygen saturation, heart rate, and activity in vaccinated and unvaccinated male and female Sprague Dawley rats using a non-invasive pulse oximetry system (Figure 5). As expected, FEN (0.1 mg/kg, SC) significantly decreased oxygen saturation (average maximum decrease to 68% at 10- and 15-m), heart rate (average maximum decrease to 231 bpm at 10-m), and activity counts to near zero levels (10–20-m) in unvaccinated male rats. These effects were completely blocked by the vaccine. FEN-induced depression of oxygen saturation and heart rate found in male rats in the present study are similar to results from a previous study that used the same procedure and dose of FEN [23]. In that study, they assessed the ability of various haptens coupled with KLH or CRM to attenuate FEN-induced respiratory depression (oxygen saturation) and bradycardia (low heart rate) in vaccinated and unvaccinated male Sprague Dawley rats. Vaccination (60 μg conjugate, 90 ug aluminum hydroxide) administered (IM) on days 0, 21, 42 and 63 with various anti-FEN conjugates protected against FEN-induced decreases in oxygen saturation and heart rate. Immunization with a FEN-based hapten conjugated to a KLH carrier protein (25 μg conjugate, 0.5 mg aluminum hydroxide) administered on days 0, 21, and 42 also blocked these physiological responses as assessed under a cumulative FEN dosing procedure [25].

Dose translation from rat to human using a formula based on body surface area and species-specific Km factor often used in cancer research indicates the FEN dose we used (0.1 mg/kg) was well above doses that cause respiratory depression and death (0.01 mg/kg or 1.13 mg/70 kg individual) [53]. Indeed, significant respiratory depression following intravenous (2.9 μg/kg) and sublingual FEN (400 μg and 800 μg) in humans are achieved with much lower doses [54,55]. Although not optimal, studies determining FEN levels following transdermal FEN administration linked to overdose death may be a better translational comparison since we administered FEN SC. Forensic toxicological analysis of FEN blood levels post-mortem and delivered using transdermal patches show that the wide range of FEN concentrations (9.1 μg–36 μg/L) were affected by multiple factors most notably, poly-drug use [56,57,58]. FEN concentrations from presumed suicide cases that had multiple FEN transdermal patches at different concentrations on their bodies show an astonishing range of 0.025–4.35 mg/h [56].

Overall, we find that vaccination with FEN-CRM+dmLT is effective in blocking or attenuating various effects of FEN in both male and female rats. However, some measures suggest it has greater effects in male rats. Rats of both sexes show robust and enduring anti-IgG antibody responses but males show greater amounts of anti-FEN IgG antibodies at weeks 8 and 10 (Figure 1B,C). Yet, we find the vaccine prevents FEN from entering the brain (Figure 2E and Figure 4E) in rats of both sexes at week 20 when there were no sex differences in anti-FEN IgG antibodies. Further, the ability of the vaccine to block the rate-disrupting effects of FEN does not differ between the sexes. The vaccine also attenuates FEN-induced decreases in heart rate and oxygen saturation as well as activity levels to a greater extent in male vs. female rats. Generally, females exhibit higher antibody responses and cell-mediated immunity but also show more frequent adverse events compared to males following vaccination for several diseases [59]. This effect is highly dependent upon the type of vaccine administered; some studies report more robust antibody response in males compared to females whereas other studies report no sex difference [60]. Nevertheless, differences in vaccine-induced antibody response may reflect several factors such as age, reproductive status, sex hormones, genetics, immune response, and environment [41,61].

Some sex differences in the efficacy of the vaccine seen in the present study are likely due to differences in baseline responses to FEN. For example, while both doses of FEN led to robust analgesic responses in unvaccinated male rats in the tail flick assay, FEN was only effective at the high dose for unvaccinated females. Similarly, FEN produced large decreases in oxygen saturation, heart rate, and activity levels in unvaccinated male rats. Except for decreases in oxygen saturation seen at a two time points, FEN did not affect these responses in unvaccinated female rats. Yet, unvaccinated female rats had higher brain levels of FEN than male rats after FEN (0.1 mg/kg) administration and there were no sex differences in FEN-induced disruptions of lever press responding across a broad dose range in the schedule-controlled responding study. We cannot conclude that the efficacy of the vaccine is lower in female rats because in some tests, the responses to FEN are quite minimal in female rats. Further, in cases in which females show clear FEN-induced responses, these are decreased by vaccination.

Prior research demonstrates that male rats are more sensitive to the antinociceptive effects of most opioids compared to females [35] except for FEN [36,37]. That we found sex differences in response to FEN in the tail flick test contrasts with findings from two studies. Peckham and Traynor (2006) assessed a broad range of FEN doses in male and female Sprague Dawley rats with the warm-water tail withdrawal (50 °C) test [37]. ED50s generated from the analgesic tests indicated no difference between males and females (0.06 mg/kg for both sexes). Additional results from that study did see sex differences with other opioids (e.g., morphine, hydrocodone, and hydromorphone) and these effects were not due to differences in drug potency, efficacy, or affinity. Another study used tail withdrawal and hot plate to test various doses of FEN (0.01, 0.03, 0.056 mg/kg) at multiple time points in Sprague Dawley rats [36]. Results showed no difference in nociception between male and female rats over time. The discrepancy between results from these studies may be due to the timing or dose of FEN administration. Results from human clinical trials assessing potential sex-dependent differences in opioid use for pain are mixed; some evidence suggests males tend to require more opioids than females because the efficacy of opioids for pain is greater in females [33,34,62]. Mechanisms that mediate these significant sex-differences in FEN-induced antinociception and on physiology and activity are unknown but may relate to several factors including: (1) differences in responses of opioid receptor subtypes; (2) type of nociception test; (3) gonadal hormones or estrus cycle effects; and (4) duration and intensity of the nociceptive stimulus [63]. Although we did not control for potential estrus effects, results from the nociception tests and physiology study are consistent with a substantial literature showing that opioids produce greater responses in male rats compared to female rats with some exceptions [37].

Low doses of opioids typically increase while higher doses decrease activity depending upon the type of test and the timing of the test after drug administration. Operant schedule-controlled responding procedures are a behavioral pharmacological tool used to evaluate the functional effects of drugs by assessing the ability of the drug to alter baseline lever press response rates after administration of a broad range of drug doses using a within-subject test procedure [64]. We used this tool to evaluate the ability of the vaccine to alter the rate-disrupting effects due to administration of higher FEN doses in rats that lever-pressed for sucrose pellets under a fixed ratio (FR15) schedule of reinforcement [44]. To test for specificity of the vaccine, we also examined response rates after MOR administration before and after vaccination. Consistent with previous studies [65,66,67], FEN dose-dependently decreased rates of responding to near zero levels at the highest dose (0.1 mg/kg). Re-testing post-vaccination completely reversed FEN’s ability to decrease response rates. This is the first study to show reversal of FEN-induced disruption in rates of responding with an anti-FEN vaccine in rats. A rhesus monkey study also utilized schedule-controlled responding procedures to test an anti-FEN vaccine (400 μg of conjugate mixed with 600 μg CpG oligodeoxynucleotide and 1 mg aluminum hydroxide) and showed a 10-fold shift in FEN-induced decreases on lever-pressing rates following vaccination indicating attenuation of FEN’s effects [68]. Although this proof-of-concept vaccine study is consistent with our work, the rigorous vaccination schedule they used (6 vaccinations over 44 weeks) precludes its potential clinical application. Taken together, results show our vaccine attenuated FEN’s rate-decreasing effects on schedule-controlled responding and that this effect was specific to FEN.

MOR administration led to an increase in rates of responding above baseline at most doses tested in the schedule-controlled responding study but was not affected by vaccination. This supports the specificity of our vaccine for FEN vs. MOR. Further, as shown in Figure 6, a series of ELISA assays using various opioid coating antigens (FEN-BSA, sufentanil-BSA, morphine-BSA, methadone-BSA, buprenorphine-BSA, oxycodone-BSA) showed that IgG antibodies generated by our vaccine were specific to FEN and cross-reacted with the FEN derivative sufentanil. This non-cross reactivity to other opioids is of primary importance since individuals that might take this vaccine may be on maintenance therapy (methadone or buprenorphine) for OUD. Results suggest post-operative pain control or chronic pain syndromes for which FEN is indicated can be treated with other opioids even in persons who use our vaccine. That antibodies generated from our vaccine also recognized sufentanil is consistent with another study assessing the efficacy of an anti-FEN vaccine showing significant affinity for FEN analogs [24].

The primary mechanism of action for all SUD vaccines is for antibodies to sequester the drug of interest in the peripheral circulation preventing brain penetration. The potent effects seen in blocking FEN-induced analgesia, disruption of schedule-controlled responding, decreases in physiological measures and activity with FEN-CRM+dmLT vaccination are consistent with this mechanism of action. We find significant decreases (~90%) in brain FEN levels in both vaccinated male (Figure 2E) and female (Figure 3E) rats compared to unvaccinated control rats following a high dose of FEN. This replicates another study in rats using the same dose and route of administration of FEN in which a 73% reduction in brain FEN post vaccination with a FEN-KLH conjugate is seen [25]. Previously, we calculated binding affinity as % IgG binding to FEN-TT in the presence of a chaotropic agent and found minimally greater antibody binding of IgG from animals vaccinated with FEN-CRM+dmLT (1 μg) compared to FEN-CRM+dmLT (0.1 μg) or FEN-CRM conjugate alone. This suggests that it is IgG antibody levels rather than differences in affinity mainly contribute towards vaccine efficacy [26]. However, the affinity of other antibody isotypes may be critical. Accumulating evidence is particularly strong for IgA given the recent finding that central peri-sinus IgA plasma cells play an important role in pathogen entrapment to prevent brain penetration [69]. While we found no correlations between %MPE and IgG or IgA antibody levels in the current study, this lack of correlation probably reflects the nearly 100% blockade of FEN’s analgesic effects and thus little variability across animals to reveal any relationships.

## 5. Conclusions

In summary, we report that our anti-FEN vaccine conjugate in combination with the adjuvant dmLT produced significant amounts of anti-FEN antibodies that were associated with complete blockade of FEN-induced analgesia in the tail flick and hotplate tests, rate-disrupting effects on schedule-controlled responding, and physiological effects. We also identified sex-dependent effects of FEN showing greater impact in male rats compared to female rats whereas our vaccine was effective in rats of both sexes. Vaccination also significantly reduced FEN entry into the brain and anti-FEN antibodies targeted FEN with no cross-reactions to other opioids. These preclinical results demonstrate efficacy in neutralizing FEN’s effects and warrant further development as a potential therapeutic for OUD and overdose in humans. We expect minimal side effects in clinical trials because the two components of our formulation (CRM and dmLT) are already in other vaccines on the market or have been tested in multiple human clinical trials and shown to be safe and effective. Further, the effective dose of dmLT used in human clinical trials is comparable to the dose used in the present study. Since low vaccine concentrations elicit adequate anti-FEN antibody levels, we expect there to be no adverse events when this vaccine is tested in humans.

## Figures and Tables

**Figure 1 pharmaceutics-14-02290-f001:**
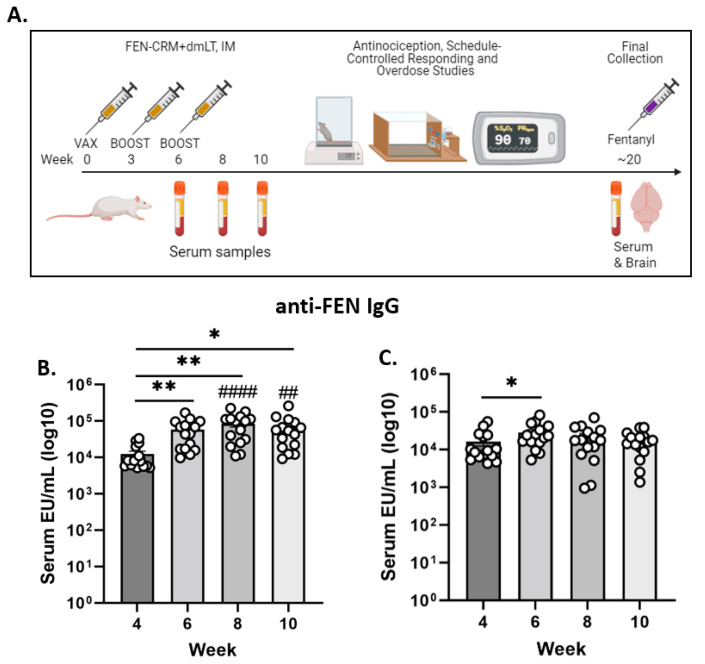
Experimental design sequence and immunogenicity of the FEN-CRM+dmLT vaccine formulation. Timeline and sequence of the experiments are presented in (**A**) and was created with BioRender.com. Rats (N = 15) were vaccinated at weeks 0, 3 and 6 and blood samples obtained at 6, 8, 10 and blood and brain samples taken at 20 weeks. IgG antibody levels were determined using ELISA with FEN-BSA as the coating antigen. Anti-FEN IgG antibody levels over weeks in male Sprague Dawley rats are presented in (**B**). Anti-FEN IgG antibody levels over weeks in female Sprague Dawley rats are shown in panel (**C**). Data are presented as serum EU (ELISA Units)/mL (log 10, mean ± SEM). * *p* > 0.05, ** *p* < 0.01; Male vs. Female ^##^
*p* < 0.01, ^####^
*p* < 0.0001.

**Figure 2 pharmaceutics-14-02290-f002:**
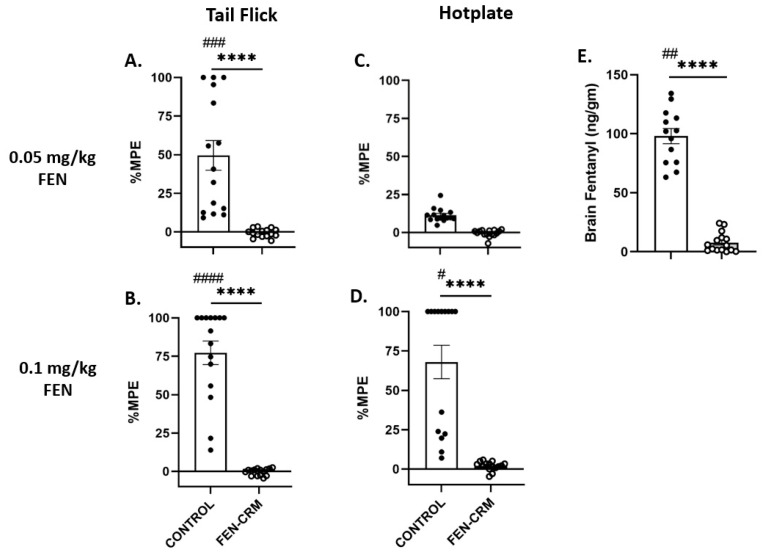
Vaccine efficacy against FEN-induced anti-nociception and FEN distribution in male Sprague Dawley rats. Rats were vaccinated with PBS (CONTROL, open bars, N = 15) or (FEN-CRM, gray bars, N = 15) at day 0, then boosted at 3 and 6 weeks. At 8–12 weeks post initial vaccination, rats were administered one of two doses of FEN (0.05 mg/kg and 0.1 mg/kg, SC) then tested in the tail flick (**A**,**B**) and hotplate (**C**,**D**) nociception assays. Data are presented as mean ± SEM %MPE (Maximal Possible Effect). Vaccinated rats showed no response to FEN-induced analgesia in the tail flick and hotplate assays whereas unvaccinated rats showed robust analgesic effects with the high FEN dose. Brain (**E**) FEN levels were obtained following administration of FEN (0.1 mg/kg, SC) at approximately week 20 post initial vaccination in the same rats. FEN was prevented from penetrating the brain in vaccinated rats but not in unvaccinated rats. **** *p* < 0.0001; CONTROL male vs. CONTROL female groups: ^#^
*p* < 0.05, ^##^
*p* < 0.01, ^###^
*p* < 0.001, ^####^
*p* < 0.0001.

**Figure 3 pharmaceutics-14-02290-f003:**
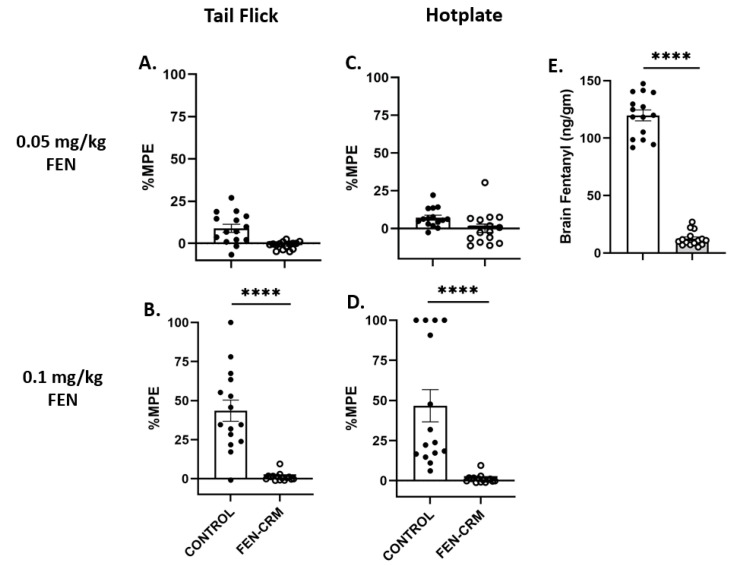
Vaccine efficacy against FEN-induced anti-nociception and FEN distribution in female Sprague Dawley rats. Rats were vaccinated with PBS (CONTROL, open bars, N = 15) or (FEN-CRM, gray bars, N = 15) at day 0, then boosted at 3 and 6 weeks. At 8–12 weeks post initial vaccination rats were administered one of two doses of FEN (0.05 mg/kg and 0.1 mg/kg) then tested in the tail flick (**A**,**B**) and hotplate (**C**,**D**) nociception assays. Data are presented as mean ± SEM %MPE (Maximal Possible Effect). Vaccinated rats showed no response to FEN-induced analgesia in the tail flick and hotplate assays whereas unvaccinated rats showed robust analgesic effects at the high FEN dose. Brain (**E**) FEN levels were obtained following administration of FEN (0.1 mg/kg, SC) at approximately week 20 post initial vaccination in the same rats that underwent analgesic testing. FEN was prevented from penetrating the brain in vaccinated rats but not unvaccinated rats. **** *p* < 0.0001.

**Figure 4 pharmaceutics-14-02290-f004:**
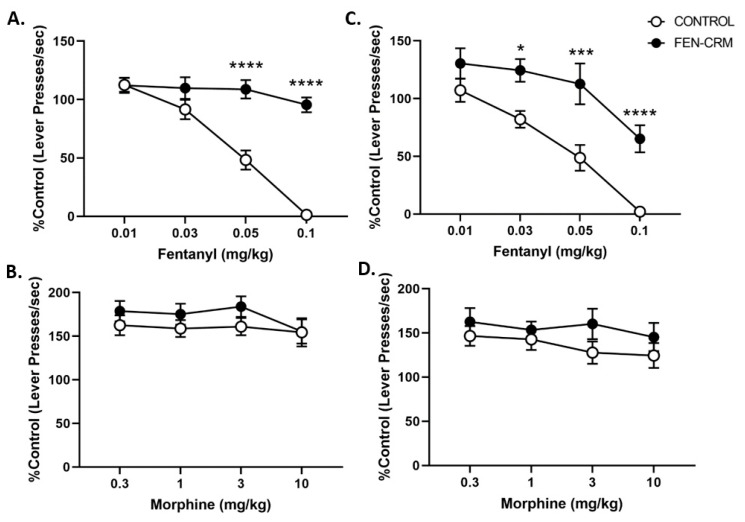
Effects of FEN and MOR on schedule-controlled responding in male and female Sprague Dawley rats. Rats (N = 13–14 males (**A**,**B**); N = 12, females (**C**,**D**)) were trained to lever press for food under an FR15 (frequency ratio) schedule of responding. Once a rat met testing criteria, it was administered one of four doses of FEN (**A**,**C**) or MOR (**B**,**D**). Rats were then vaccinated at 0, 3 and 6 weeks and retested with both compounds between weeks 8–10 post-initial vaccination. Data are presented as mean ± SEM % control rate of responding. Vaccination significantly blocked the rate decreasing effects of FEN but did not alter the influence of MOR on response rates in both male and female rats. * *p* < 0.05, *** *p* < 0.001, **** *p* < 0.0001.

**Figure 5 pharmaceutics-14-02290-f005:**
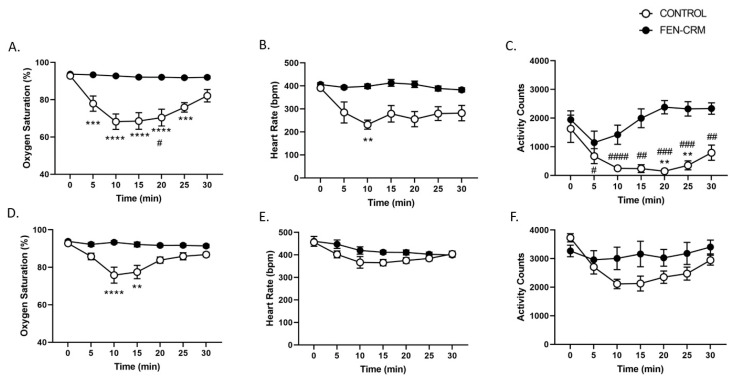
Vaccine efficacy against FEN-induced physiological effects in male and female Sprague Dawley rats. Sprague Dawley rats (male, CONTROL, N = 6, FEN-CRM, N = 8, (**A**–**C**); female, CONTROL, N = 9, FEN-CRM, N = 6, (**D**–**F**)). Groups were vaccinated with PBS or FEN-CRM+dmLT (day 0, 3 and 6 weeks, IM) and at approximately 10 weeks post initial vaccination rats were administered 0.1 mg/kg FEN (SC) and oxygen saturation (%), heart rate (beats per minute, bpm), and activity (counts) were measured using a pulse oximetry system. FEN significantly decreased all measures in unvaccinated male rats and these effects were attenuated by the vaccine. FEN significantly decreased oxygen saturation in unvaccinated female rats that was reversed by vaccination. CONTROL vs. FEN-CRM male and female groups: ** *p* < 0.01, *** *p* < 0.001,**** *p* < 0.0001; CONTROL male vs. CONTROL female groups: ^#^
*p* < 0.05, ^##^
*p* < 0.01, ^###^
*p* < 0.001, ^####^
*p* < 0.0001.

**Figure 6 pharmaceutics-14-02290-f006:**
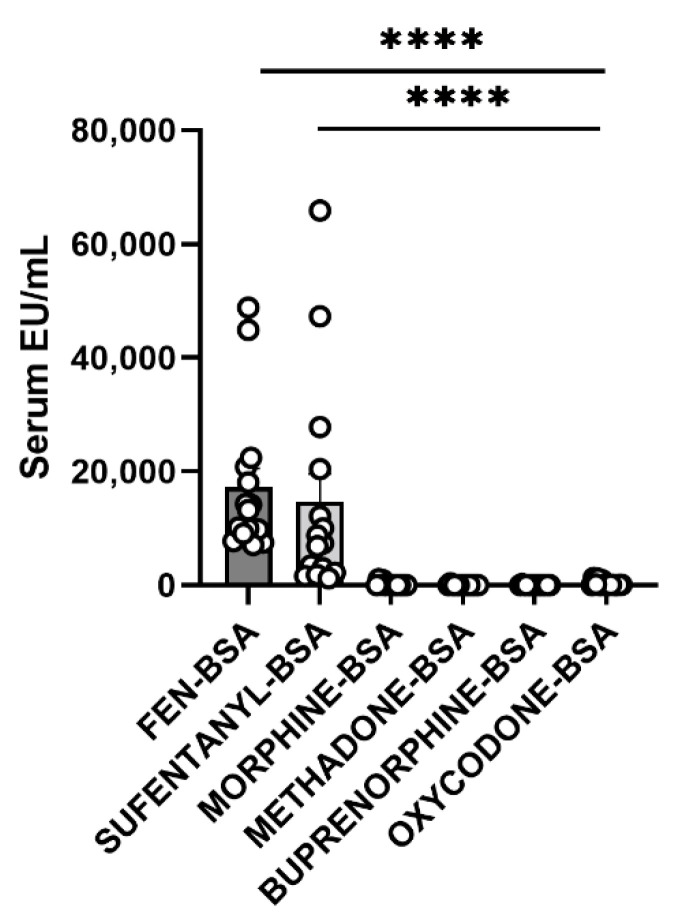
Anti-FEN antibody specificity in samples from vaccinated rats. Specificity of anti-FEN IgG antibodies generated by FEN-CRM+dmLT was determined using ELISA by coating the plate with different target antigens and processing serum collected at week 20 from unvaccinated control and vaccinated male rats. As shown, anti-FEN antibodies bound only to the FEN-BSA and SUFENTANIL-BSA antigens and not to other opioids tested. Data are presented as Serum EU (ELISA Units)/mL subtracted from control. **** *p* < 0.0001.

## Data Availability

Not applicable.

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
