# Peer review of "An Immunconjugate Vaccine Alters Distribution and Reduces the Antinociceptive, Behavioral and Physiological Effects of Fentanyl in Male and Female Rats"

_pharmaceutics, 2022, doi:10.3390/pharmaceutics14112290_

Round 1

Reviewer 1 Report

The manuscript by Haile et al. is well written and describes a well designed and executed study of a fentanyl vaccine in rats. The study observes sex differences in both fentanyl challenge and vaccine induced immune responses and efficacy from fentanyl challenge.

The study lacks one significant experiment. The competition ELISA study in Fig 6 should also include naloxone and naltrexone, which are significant therapeutics for substance abuse. 

Author Response

Reviewer 1

The manuscript by Haile et al. is well written and describes a well designed and executed study of a fentanyl vaccine in rats. The study observes sex differences in both fentanyl challenge and vaccine induced immune responses and efficacy from fentanyl challenge.

The study lacks one significant experiment. The competition ELISA study in Fig 6 should also include naloxone and naltrexone, which are significant therapeutics for substance abuse.

Response:

Thank you for your comment. Please note that we have added new ELISA data to the cross reactivity study presented in Fig 6 from plates coated with SUFENTANIL-BSA.  Details of the synthesis and haptinization ratio have been added to Section “2.3 Antigens and adjuvants”.  The Results and Discussion sections have also been updated to reflect these additional findings. We have also made a spelling correction (hydroxycodone>oxycodone).

We used an indirect ELISA assay whereby we coat the plate with different antigens (as shown in Fig 6) then add serum from vaccinated animals, then add a secondary conjugate antibody (anti-IgG+alkaline phosphatase). We show that anti-fentanyl IgG antibodies generated by our vaccine bind to the FEN-BSA hapten and SUFENTANIL-BSA hapten but not to other opioids showing specificity of the antibodies. Description of the synthesis and characterization of SUFENTANIL-BSA is under “Supplementary Material”.

You are correct that naltrexone is indicated for the treatment of OUD and naloxone is used to address opioid overdose.  It is highly unlikely however, that anti-fentanyl antibodies would bind to naltrexone or naloxone because these compounds have no structural similarities with fentanyl.  Indeed, antibodies generated from a different anti-fentanyl vaccine did not bind to either naltrexone or naloxone (Barrientos et al. Molecular Pharmaceutics, 2020). Further, another anti-fentanyl vaccine in combination with naloxone completely blocked fentanyl-induced decreases in oxygen saturation whereas naloxone alone did not (Raleigh et al. Journal of Pharmacology and Experimental Therapeutics, 2019).  Taken together, it appears that anti-fentanyl antibodies do not recognize naltrexone or naloxone and probably have an enhanced effect in blocking fentanyl-induced effects when combined with an anti-fentanyl vaccine.  We expect our anti-fentanyl vaccine to have these same characteristics.

Reviewer 2 Report

In the manuscript submitted by Colin N Haile et al, the authors studied the immunological and antinociception efficacy of the Fentanyl (FEN) vaccine which was prepared by combining FEN-CRS conjugates with dmLT as adjuvant reported by the authors and others previously. The authors tested the FEN vaccine in both male and female Sprague Dawley rats. The results of the immunogenicity, behavioral tests of nociception, physiological effects and the level of brain fentanyl of both male and female rats were reported and compared. The authors conclude that the vaccine is effective on both male and female rats. Some sex related effects to the vaccine were found and discussed. This work may provide useful information for further development of this FEN vaccine as potential therapeutic for humans.

Some specific comments as below:

1>     Did the author test the IgA levels of both male and female rats after the immunization? Please comment on this since the dmLT is involved in the mucosal immune responses, and the authors’ previous work on mice implied the correlation of the protection of such vaccine with anti-FEN IgA.

2>      Please add the information of the haptenization ratio of FEN-CRM, since this number is important for the immunogenicity of the conjugates.

3>     Similarly, for Figure 6, please provide the information about the number (level) of small drug molecules on each BSA and the conjugation chemistry.

4>     Please add some more details about the preparation of brain samples for FEN quantification and how to you normalize the amount of sample from different rats for Elisa assay.

5>      Line 169, is the super pure water ddH2O?

Author Response

In the manuscript submitted by Colin N Haile et al, the authors studied the immunological and antinociception efficacy of the Fentanyl (FEN) vaccine which was prepared by combining FEN-CRS conjugates with dmLT as adjuvant reported by the authors and others previously. The authors tested the FEN vaccine in both male and female Sprague Dawley rats. The results of the immunogenicity, behavioral tests of nociception, physiological effects and the level of brain fentanyl of both male and female rats were reported and compared. The authors conclude that the vaccine is effective on both male and female rats. Some sex related effects to the vaccine were found and discussed. This work may provide useful information for further development of this FEN vaccine as potential therapeutic for humans.

Some specific comments as below:

  • Did the author test the IgA levels of both male and female rats after the immunization? Please comment on this since the dmLT is involved in the mucosal immune responses, and the authors’ previous work on mice implied the correlation of the protection of such vaccine with anti-FEN IgA.

We appreciate the Reviewer pointing out this important finding in our previous mouse study. We did quantify IgA levels following vaccination in the present study however; they did not correlate with tail flick or hot plate latencies.  A major difference between the rat and mouse studies is that we observed nearly 100% blockade of the analgesic effects of fentanyl which does not leave enough variability in the data to reveal any potential correlations.  We have added the following statement at the end of the Discussion for clarity and to address this difference between our studies.

While we found no correlations between %MPE and IgG or IgA antibody levels (data not shown) in the current study, this lack of correlation probably reflects the nearly 100% blockade of FEN’s analgesic effects and thus little variability across animals to reveal any relationships.”

  • Please add the information of the haptenization ratio of FEN-CRM, since this number is important for the immunogenicity of the conjugates.

Agree. We added the following sentence to section 2.3 Antigens and adjuvants.   

“FEN-CRM conjugate characterization by MALDI-TOF gave a haptenization ratio of 2.4-2.6.”

  • Similarly, for Figure 6, please provide the information about the number (level) of small drug molecules on each BSA and the conjugation chemistry.

Please note that we have added new ELISA data to the cross-reactivity study presented in Figure 6 from plates coated with SUFENTANIL-BSA.  Details of the synthesis and haptinization ratio have been added to Section “2.3 Antigens and adjuvants”.  The Results and Discussion sections have also been updated to reflect these additional findings. We have also made a spelling correction (hydroxycodone>oxycodone). Description of the synthesis and characterization of SUFENTANIL-BSA is under “Supplementary Material”.

The other conjugates used in the cross-reactivity study were purchased from a company (CalBioreagents, Foster City, CA) mentioned in the “2. Materials and Methods” section.  We contacted the company to get the information requested by the Reviewer and they refused based on the fact that their conjugation chemistry process is proprietary.

  • Please add some more details about the preparation of brain samples for FEN quantification and how to you normalize the amount of sample from different rats for Elisa assay.

We thank the Reviewer for noting the lack of detail in processing the brain samples.  The brain samples were processed and assayed for protein concentrations.  Equal amounts of brain protein were assayed for each rat to normalize the amount of sample across animals.  We have added the following details below to section “2.5 Antibody Levels, Cross-reactivity and Fentanyl Quantification.”

“Brains were first homogenized in super pure water then underwent centrifugation (10,000 rpm, 10 min) and supernatant removed. Protein quantification of each supernatant from each animal was obtained using BCA protein assay kit (ThermoFisher Scientific). Equal protein concentrations from each rat were assayed for FEN concentration.”

  • Line 169, is the super pure water ddH2O?

We use the PURELAB Ultra water purification system. Super Pure Water (or Ultra Pure) “is water that has been purified to high levels of specification. As a standard, the water contains only H20, as well as balanced number of H+ and OH- ions. It has a resistivity of 18.2 MΩ.cm, TOC < 10 ppb and bacterial count <10 CFU/ml. To be classified as ultrapure, water must not contain any detectable endotoxins. This level of purity makes it a perfect reagent for laboratory work.”

https://www.elgalabwater.com/ultrapure-water